# LightSeq: Sequence Level Parallelism for Distributed Training of Long Context Transformers

## Abstract

Increasing the context length of large language models (LLMs) unlocks fundamentally new capabilities, but also significantly increases the memory footprints of training. Previous model-parallel systems such as Megatron-LM partition and compute different attention heads in parallel, resulting in large communication volumes, so they cannot scale beyond the number of attention heads, thereby hindering its adoption. In this paper, we introduce a new approach, LightSeq, for long-context LLMs training. LightSeq has many notable advantages. First, LightSeq partitions over the sequence dimension, hence is agnostic to model architectures and readily applicable for models with varying numbers of attention heads, such as Multi-Head, Multi-Query and Grouped-Query attention. Second, LightSeq not only requires up to 4.7× less communication than Megatron-LM on popular LLMs but also overlaps the communication with computation. To further reduce the training time, LightSeq features a novel gradient checkpointing scheme to bypass an forward computation for memory-efficient attention. We evaluate LightSeq on Llama-7B and its variants with sequence lengths from 32K to 512K. Through comprehensive experiments on single and cross-node training, we show that LightSeq achieves up to 1.24-2.01× end-to-end speedup, and a 2-8× longer sequence length on models with fewer heads, compared to Megatron-LM. Anonymous codes available at `https://anonymous.4open.science/r/lightseq-anonymized`.

## 1 Introduction

Transformers with long-context capabilities have enabled fundamentally new applications, such as comprehensive document understanding, generating a complete codebase, and extended interactive chatting (Osika, 2023; Liu et al., 2023; Li et al., 2023). However, training LLMs with long sequences induces large activation memory footprints, posing new challenges to existing distributed systems.

One effective method for reducing these large activation memory footprints is to partition the activation across devices. To achieve this, existing systems like Megatron-LM (Korthikanti et al., 2023; Shoeybi et al., 2019) usually partition the attention heads. However, this design poses a strong assumption that the number of attention heads must be divisible by the parallelism degree, which does not hold for many model architectures. For example, Llama-33B (Touvron et al., 2023) and its fine-tuned versions (e.g., Tulu-30B (Wang et al., 2023)) have 52 attention heads, Falcon-7B (Penedo et al., 2023) has 71 attention heads, and GPT-2-XL (Radford et al., 2019) has 25 attention heads. These numbers are not divisible by commonly chosen parallelism degrees such as 8, 16, and 32, according to the topology of NVIDIA clusters. In addition, partitioning attention heads restricts the maximum parallelism degree to be no greater than the number of attention heads. However, many popular LLMs do not have enough attention heads for it to scale up, e.g., CodeGen (Nijkamp et al., 2022) only has 16 attention heads. Moreover, many works have shown that the future Transformer architecture design may have even fewer attention heads. For example, Bian et al. (2021) demonstrates that Transformers with a single head outperforms its multi-head counterparts, representing a challenging scenario for solutions like Megatron-LM.

To scale beyond the number of heads, we propose partitioning solely the input tokens (i.e., sequence parallelism) rather than the attention heads, along the line of research in sequence parallelism Li et al. (2021); Korthikanti et al. (2023). We present a solution that is agnostic to the model architecture and exhibits a maximal parallelism degree that scales with the sequence length. Specifically, we introduce a parallelizable and memory-efficient exact attention mechanism, DISTATTN, in (§3.1). Our design enables opportunities for overlapping, where we can hide communication into attention computation(§ 3.2). We also propose a load-balancing technique to avoid the computation bubble caused by the unbalanced workload in causal language modeling (§3.2). While extending the FlashAttention (Dao, 2023) algorithm to DISTATTN, we found a way to leverage the underlying rematerialization logic to significantly improve the speed of gradient checkpointing training (§ 3.3). This technique also applies to non-distributed usage of memory-efficient attention, and in our experiments translates to an additional 1.31× speedup (§ 4.3).

Our main contributions are:

1. We design LIGHTSEQ, a long-context LLM training prototype based on sequence-level parallelism. We develop a distributed memory-efficient exact attention DISTATTN, with novel load balancing and communication overlapping scheduling for causal language modeling.

2. We propose a novel checkpointing strategy that bypasses one attention forward pass when using memory-efficient attention with gradient checkpointing training.

3. We evaluate LIGHTSEQ on Llama-7B and its variants with different attention heads patterns, and demonstrate up to 2.01× end-to-end speedup compared to Megatron-LM in long-context training. We further show that LIGHTSEQ scales beyond the number of attention heads and enables 2-8× longer sequences training.

## 2 RELATED WORK

**Memory-efficient attention.**    Dao et al. (2022) and Lefaudeux et al. (2022) propose to use an online normalizer (Milakov & Gimelshein, 2018) to compute the attention in a blockwise and memory-efficient way. It reduces peak memory usage by not materializing large intermediate states, e.g. the attention matrix or the up projection matrix output of the MLP layers (Liu & Abbeel, 2023). Instead, the attentions are computed in smaller blocks and only the final activation are stored. In the backward pass, the intermediate states need to be recomputed. Research on sparse attention computes only a sparse subset of the attention score, which also reduces the memory footprints yet may lead to inferior performance (Beltagy et al., 2020; Sun et al., 2022; Zaheer et al., 2020). In this work, we limit our scope to exact attention.

**Sequence parallelism, model parallelism, and FSDP.**    Li et al. (2021) is among the first to parallelize along the sequence dimension. However, it is not optimized for the computational pattern of causal language modeling and is incompatible with memory-efficient attention, which are crucial to long-context LLM training. Model parallelism partitions model parameters and also distributes the activation in parallel LLM training. Megatron-LM (Korthikanti et al., 2023) proposes a hybrid usage of tensor parallelism and sequence parallelism to better reduce the activation on a single device and is the main baseline of the paper. Fully sharded data-parallelism (FSDP) (Zhao et al., 2023; Rajbhandari et al., 2020) distributes optimizer states, gradients, and model parameters onto different devices and gathers them on-the-fly. It is orthogonal to our work, and we use LIGHTSEQ in tandem with FSDP to further reduce memory acquired by models in experiments.

**Pipeline parallelism**    Pipeline parallelism Korthikanti et al. (2023) also partitions the activation. However, it keeps a high memory pressure to the first stage when applying interleaved pipeline parallelism to minimize the computation bubble Korthikanti et al. (2023). We show in § 4.2 that pipeline parallelism is less effective in supporting long sequence lengths compared with tensor model parallelism and our sequence parallelism. Thus, we focus on comparing with tensor model parallelism (combined with sequence parallelism) in this work and only consider including pipeline parallelism for comparison when the tensor parallelism is limited by the number of heads.

**Gradient checkpointing.**    Gradient checkpointing (Chen et al., 2016) trades computation for memory by not storing the activation for certain layers and recomputing their activations during forward. Selective checkpointing (Korthikanti et al., 2023) proposes to only recompute the attention module as it requires large memory but with small FLOPs (in smaller context length). Checkmate (Jain et al., 2020) searches optimal checkpointing using integer linear programming. However,

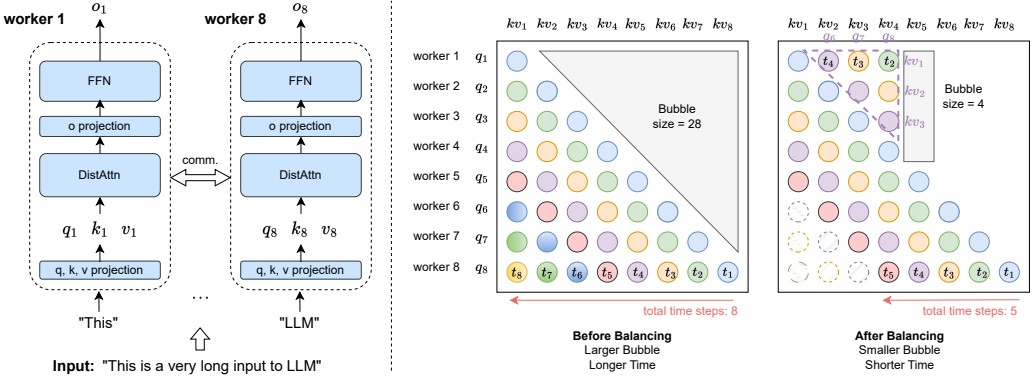

Figure 1: Left: Sequence parallelism in LIGHTSEQ. The input sequence is split into chunks along the sequence dimension and distributed to different workers (8 workers in the illustration). During forward and backward, only the attention module, DISTATTN, requires communication of intermediate tensors like $k$ and $v$. Some modules like LayerNorm are ignored for simplicity. Right: Illustration of the load-balanced scheduling. Each circle is a unit of computation. And circles in the same color means that they are computed in the same time step. For instance, the rightmost and bottommost circle means that at time step 1 (t1), worker 8 is executing $attn(q_8, k_8, v_8)$. Similarly, green color denotes computations that happen at the second time step ($t_2$). At $t_2$, worker 1 is executing $attn(q_8, k_1, v_1)$. "Bubble size" represents the times that a worker is idle. Causal language modeling naturally introduces imbalanced workloads, e.g., worker 1 is idle from time step 2 to time step 8 before balancing. We reduce the bubble fraction by allocating computation from the busy worker (e.g., worker 8) to the idle worker (e.g., worker 1), so worker 1 is only idle at time step 5 after balancing. A more detailed illustration of the load-balancing design can be found is Appendix C.

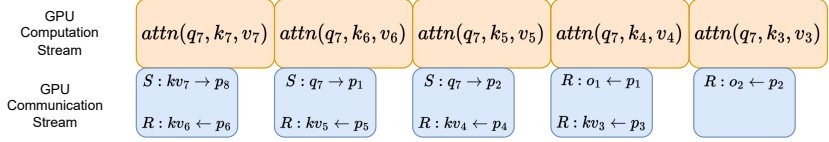

Figure 2: Forward pass example of overlapping communication using worker 7 out of 8 workers. $o$ denotes the attention output computed by a remote worker. For instance, $o_1 = attn(q_7, k_1, v_1)$ for worker 7. In the communication stream, "S" stands for sending, and "R" stands for receiving. For instance, $S : kv_7 \rightarrow p_8$ denotes sending the local $kv_7$ to the remote worker $p_8$. In Appendix C, the communication schema at each time step is reflected in Figure 7.

none of these designs have considered memory-efficient attention kernels which perform recomputation inside the computational kernel to avoid materializing large tensors. As a result, many previous recomputation policies become less effective. In this work, we focus on checkpointing at the boundary of every transformer layer, which is a popular strategy adopted by many current open-sourced projects such as FastChat (Zheng et al., 2023).

## 3   METHOD

In this section, we describe the design of the key components in LIGHTSEQ. We first introduce a distributed memory-efficient attention, DISTATTN (§3.1) which parallelizes the computation along the sequence dimension. We then introduce a load-balanced scheduling for causal language modeling to reduce the computation bubble as well as an asynchronous communication design that overlaps the communication into computation (§3.2). Finally, we propose a rematerialization-aware checkpointing strategy (§3.3) which effectively cuts off the recomputation time in gradient checkpointing.

### 3.1   DISTATTN: DISTRIBUTED MEMORY-EFFICIENT ATTENTION

The core idea in DISTATTN is to split the input sequence consisting of $N$ tokens evenly across $P$ workers (e.g. GPUs) along the sequence dimension. Each worker is therefore responsible for com-

puting the forward and backward pass for only $N/P$ of the $N$ tokens. For modules like the Feed Forward Layer (FFN), Layer Norm (LN), and the embedding layer the tokens can be computed independently without coordination (embarrassingly parallel) and the work is balanced across workers.

Unfortunately, for the attention modules where local tokens may need to attend to remote tokens, coordination is required. To address this, each worker collects all the keys and values associated with other tokens and then locally computes the attention following Dao (2023). To address the memory pressure introduced by collecting all other keys and values, this process is done online by streaming the key and values from workers with earlier tokens to workers with later tokens. More formally, denote $\mathbf{q}_p$, $\mathbf{k}_p$, $\mathbf{v}_p$ as the query, key, value inputs held on the $p$-th worker ($p = \{1, \cdots, P\}$), denote $attn(\mathbf{q}_p, \mathbf{k}_{p'}, \mathbf{v}_{p'})$ as the attention computation w.r.t. $p$-th chunk of the query and $p'$-th chunk of the key and value, denote $p_{\text{local}} \in \{1, \cdots, P\}$ as the local rank, and denote $p_{\text{remote}} \in \{1, \cdots, P\}$ as one of the remote ranks. Figure. 1 ("Before Balancing") shows the vanilla version of DISTATTN, where each worker computes the attention for $\mathbf{q}_{p_{\text{local}}}$ and loops over both the local and the remote key and value blocks. We fetch $\mathbf{k}_{p_{\text{remote}}}$ and $\mathbf{v}_{p_{\text{remote}}}$ from rank $p_{\text{remote}}$ before the computation of $attn(\mathbf{q}_{p_{\text{local}}}, \mathbf{k}_{p_{\text{remote}}}, \mathbf{v}_{p_{\text{remote}}})$. In Appendix. A, we provide pseudo-code on how to use DISTATTN in LIGHTSEQ, on the $p$-th worker where there are $P$ total workers.

### 3.2 LOAD BALANCED SCHEDULING WITH COMMUNICATION AND COMPUTATION OVERLAP

**Load balanced scheduling.** Causal language modeling objective (Brown et al., 2020; Touvron et al., 2023) is one of the most prevalent objectives for LLMs, where each token only attends to its previous tokens. This naturally introduces a work imbalance between workers in our block-wise attention: as shown in Figure 1 ("Before Balancing"), in an 8-worker ($P = 8$) scenario, the last worker needs to attend to tokens on all other 7 workers, while the first worker is idle after attending to its local tokens, which results in a total idle time of 28. In a general form, the idle fraction is $\frac{P^2 - P}{2P^2}$ ($\to \frac{1}{2}$ when $P \to \infty$), which means roughly half of the workers are idle. To reduce this idle time (a.k.a., the bubble time), we let early workers that have finished their computation for local $\mathbf{q}_{p_{\text{local}}}$ to help compute for $\mathbf{q}_{p_{\text{remote}}}$ of the later workers. For instance, we let worker 1 compute $attn(\mathbf{q}_8, \mathbf{k}_1, \mathbf{v}_1)$ and send the result to worker 8. When the number of workers is odd, the idle fraction is 0. When the number of workers is even, the idle fraction is $\frac{1}{2P}$, which is asymptotically 0 when scaling to more number of workers. We detail the load-balancing design in Appendix C.

**Communication and computation overlap.** DISTATTN relies on peer-to-peer (P2P) communication to fetch the $\mathbf{k}$, $\mathbf{v}$ (or $\mathbf{q}$ chunks in the load balanced scheduling) from remote devices before computing the corresponding attention block. However, these communications can be easily overlapped with the computation of the former blocks. For instance, When the first worker is computing attention for its local token, it can pre-fetch the next chunk of tokens it needs for the next time step. In modern accelerators, this can be done by placing the attention computation kernel in the main GPU stream, and the P2P communication kernel in another stream, where they can run in parallel (Zhao et al., 2023). We demonstrate the overlapped scheduling for worker 7 on the 8 workers example in Figure. 2. Empirically, we find this optimization greatly reduces the communication overhead (§4.3).

### 3.3 REMATERIALIZATION-AWARE CHECKPOINTING STRATEGY

The de-facto way of training transformers requires gradient checkpointing. Often, the system uses heuristics to insert gradient checkpoints at each Transformer layer (Wolf et al., 2019). However, with the presence of Dao et al. (2022), we found the previous gradient checkpointing strategy will cause an extra recomputation of the flash attention forward kernel. Concretely, when computing the gradient of the MLP layer, Wolf et al. (2019) will recompute the forward of the entire Transformer layer, including the one in flash attention. However, when computing the gradient of the flash

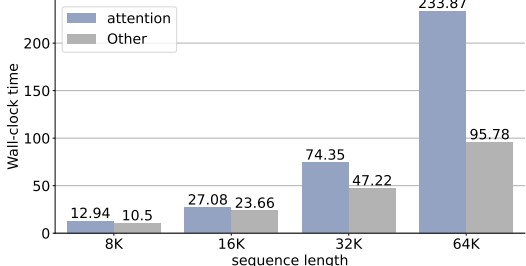

Figure 3: Time breakdown of attention versus other modules in a forward pass. Time measured with Flash-Attention (Dao, 2023) on a single 40GB A100 GPU. (Unit ms).

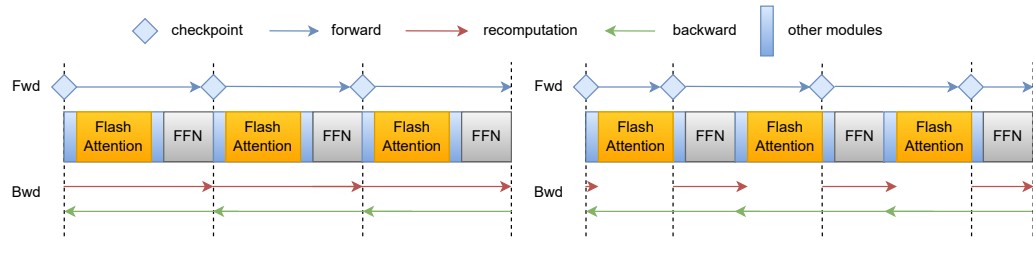

Figure 4: Comparison of HuggingFace gradient checkpointing strategy and our materialization-aware gradient checkpointing strategy. Note that our checkpointing strategy **saves an entire flash attention forward per layer** in recomputation.

attention kernel, it needs to re-compute the forward of the flash attention again. Essentially, this is because flash attention will not materialize the intermediate values during the forward, and will re-compute it during the backward, regardless of the re-computation strategy in the outer system level. To tackle this, we propose to insert checkpoints at the output of the flash attention kernel, instead of at the Transformer layer boundary. In this case, we only need to recompute the forward of flash attention once, effectively saving a forward of attention for each Transformer layer as shown in Figure. 4. In Figure. 3, we show the attention time dominates in the forward pass when scaling up the sequence length, which indicates our method can save $\sim 0.23 \times 32$ (i.e., $\sim 7$) seconds when training a 64K sequence example on Llama-7b using the local version of flash attention. In addition, this saves a communication brought by our DISTATTN forward in the distributed training scenario. We benchmark the end-to-end speedup brought by this materialization-aware checkpointing strategy in §4.3.

**Communication and memory analysis** Denote the hidden dimension as $d$. In DISTATTN, every worker needs to fetch key and value chunks both of size $\frac{N}{P}d$ before performing the corresponding chunk-wise computation. Thus, the total communication volume in the $P$-workers system is $2 \times \frac{N}{P}d \times P = 2Nd$. With the causal language objective, half of the keys and values do not need to be attended, halving the forward communication volume to $Nd$. In the backward pass, DISTATTN needs to communicate keys, values, and their gradients, which has $2Nd$ volume. It adds up to $3Nd$ as the total communication volume for DISTATTN. In Megatron-LM (Korthikanti et al., 2023), each worker needs to perform six all-gather and four reduce-scatter on a $\frac{N}{P}d$ size tensor, thus giving a total communication volume of $10Nd$. Considering gradient check-pointing, Megatron-LM will perform communication in the forward again, giving a total volume of $14Nd$. On the other hand, our communication volume remains $3Nd$ because of the rematerialization-aware strategy. In conclusion, LIGHTSEQ achieves 4.7x communication volume reduction compared with Megatron-LM.

In large model training, we usually utilize techniques such as FSDP to also reduce the memory consumed by model weights. In this case, We note that the communication introduced by FSDP is only proportional to the size of model weights, which does not scale up with long sequence length. We show the end-to-end speedup with FSDP in Table 1. For clarity, we also note that LIGHTSEQ is orthogonal to FSDP and by default can be used by itself. In the situations where the model uses MQA or GQA, LIGHTSEQ further saves the communication volumes by the shared key and values, which we discuss in detail in § 4.1. However, we also note that this is a theoretical analysis, where the wall-clock time may differ because of factors such as implementations. In the experiment section, we provide wall-clock end-to-end results for comparison.

## 4 EXPERIMENTS

In this section, we evaluate LIGHTSEQ against Megatron-LM (Korthikanti et al., 2023) and show:

1. LIGHTSEQ has faster training speed on a wide range of models. It achieves up to $2.01\times$ speedup over Megatron-LM on various MHA and GQA models.

2. LIGHTSEQ supports longer sequence length by scaling beyond the number of attention heads. We show our method can support 2x-8x longer sequences than Megatron-LM.

In the ablation study, we provide the gain from each component of LIGHTSEQ: Load balancing, computation-communication overlapping, and rematerialization-aware checkpointing.

**Cluster setup.** We evaluate our method and the baseline in (1) A single A100 DGX box with 8x80 GB GPUs. These GPUs are connected with NVLink; (2) 2 DGX boxes with the same setting. These two boxes are interconnected by 100 Gbps Infiniband. This is representative of cross-node training, where the communication overhead has a larger effect. This is the default setup unless otherwise stated. (3) Our in-house cluster with 2x8 A100 40GB GPUs without Inifiniband. To save computational budget, we report some results on this cluster where conclusions can be drawn from a single-node setup or without involving cross-node training time.

**Model setup.** We evaluate our system on Llama-7B and its variants of different representative families: (1) Multi-head attention(MHA) models: LLama-7B with 4096 hidden size and 32 query(key and value) heads (Touvron et al., 2023); (2) Grouped-Query attention (GQA) models: Llama-GQA Ainslie et al. (2023), same as Llama-7B but with 8 key and value heads. During attention computation, it will first replicate to 32 heads in order to do matrix multiplication with the correct shape. (3) models with more general number of attention heads: Llama-33H. Llama-33H has the same configuration as Llama-7B but with 33 query (key and value) attention heads per layer. (4) models with fewer attention heads: we design Llama-16H, Llama-8H, Llama-4H, Llama-2H with 16, 8, 4, and 2 heads. According to Liu et al. (2021), we keep the number of attention heads by scaling the number of layers properly[1] and keep the intermediate FFN layer size the same to make the model sizes still comparable. For example, Llama-16H has 16 attention heads per layer, a hidden size of 2048, an FFN layer of size 11008, and 64 layers.

**Implementation.** LIGHTSEQ is a lightweight scheduling level prototype. In particular, we implement the load balancing and overlapping in Python and NCCL Pytorch bindings in 1000 lines of codes (Paszke et al., 2019; Jeaugey, 2017), and the checkpointing strategy in 600 lines of Pytorch. It is attention backend agnostic. To reduce the memory consumption and reach faster speed in the attention module, we use the FlashAttention2 algorithm (Dao, 2023). We use the triton (Tillet et al., 2019) implementation and minimally modify it to keep around statistics in the flash attention algorithm. We tweak all block sizes to 128 and the number of stages to 1 for the best performance in our cluster. We reuse the C++ backward kernels of FlashAttention2 because we do not need to modify the backward logic. We run LIGHTSEQ using FSDP (inter-node if applicable) so that it consumes similar memory than the Megatron-LM baseline for a fair comparison (Zhao et al., 2023). For fair comparisons, we run all comparisons using the same attention backend. We also add support for Megatron-LM so that comparing with them can produce a more insightful analysis: (1) not materializing the causal attention mask, greatly reducing the memory footprint. For instance, without this support, Megatron-LM will run out of memory with Llama-7B at a sequence length of 16K per GPU. (2) head padding where the attention heads cannot be divided by device number. All results are gathered with Adam optimizer, 10 iterations of warm-up, and averaged over the additional 10 iterations.

## 4.1 FASTER TRAINING SPEED AND BETTER SUPPORT FOR DIFFERENT MODEL ARCHITECTURES

In this section, we compare our method with Megatron-LM on three settings: (1) the multi-head attention (MHA) models where the number of key and value heads equals the number of query heads; (2) the grouped-query attention (GQA) models where the number of key and value heads is less than the number of query heads; (3) the models with arbitrary numbers of heads, i.e. the number heads is unnecessarily a multiple of the parallelism degree.

**Multi-head attention (MHA).** On the Llama-7B model, our method achieves **1.24×** and **1.44×** speedup compared to Megatron-LM in single node and cross node setting, up to the longest sequence length we experiment. This is a joint result of our overlapping communication technique and our rematerialization-aware checkpointing strategy. We analyze how much each factor contributes to this result in the ablation study ( § 4.3). We do note that our method does not achieve better performance in shorter sequences, such as per GPU 4K setting for cross node. This is because the communication dominates the training run-time, where our overlapping technique has not been able to reduce much. We leave the optimization of P2P communication on MHA models and shorter sequence length as an exciting future work.

---

[1] For instance, Llama-7B has 32 attention heads and 32 layers, thus Llama-16H has 16 attention heads per layers and 64 layers

Table 1: Per iteration wall-clock time of LIGHTSEQ and Megatron-LM (Korthikanti et al., 2023) (Unit: seconds). Speedup in bold denotes the better of the two systems in the same configuration. Time measured with 2 DGX boxes.

| Method | # GPUs | Sequence Length | | Llama-7B | | Llama-GQA | | Llama-33H | |
|---|---|---|---|---|---|---|---|---|---|
| | | Per GPU | Total | Time | speedup | Time | speedup | Time | speedup |
| Megatron-LM | 1x8 | 4K | 32K | 2.54 | 1.0x | 2.43 | 1.0x | 3.15 | 1.0x |
| | 1x8 | 8K | 64K | 6.81 | 1.0x | 6.60 | 1.0x | 8.37 | 1.0x |
| | 1x8 | 16K | 128K | 20.93 | 1.0x | 20.53 | 1.0x | 25.75 | 1.0x |
| | 1x8 | 32K | 256K | 72.75 | 1.0x | 71.93 | 1.0x | 90.21 | 1.0x |
| LIGHTSEQ | 1x8 | 4K | 32K | 2.50 | **1.02x** | 2.30 | **1.06x** | 2.58 | **1.22x** |
| | 1x8 | 8K | 64K | 5.98 | **1.14x** | 5.61 | **1.18x** | 6.08 | **1.38x** |
| | 1x8 | 16K | 128K | 17.26 | **1.21x** | 16.86 | **1.22x** | 17.77 | **1.45x** |
| | 1x8 | 32K | 256K | 58.46 | **1.24x** | 57.01 | **1.26x** | 59.96 | **1.50x** |
| Megatron-LM | 2x8 | 4K | 64K | 5.29 | 1.0x | 5.26 | 1.0x | 7.52 | 1.0x |
| | 2x8 | 8K | 128K | 14.26 | 1.0x | 14.21 | 1.0x | 20.63 | 1.0x |
| | 2x8 | 16K | 256K | 43.44 | 1.0x | 43.20 | 1.0x | 62.78 | 1.0x |
| | 2x8 | 32K | 512K | 147.06 | 1.0x | 146.38 | 1.0x | 216.70 | 1.0x |
| LIGHTSEQ | 2x8 | 4K | 64K | 6.85 | 0.77x | 4.92 | **1.07x** | 7.03 | **1.07x** |
| | 2x8 | 8K | 128K | 12.75 | **1.12x** | 9.74 | **1.46x** | 13.12 | **1.57x** |
| | 2x8 | 16K | 256K | 30.21 | **1.44x** | 28.49 | **1.52x** | 31.33 | **2.00x** |
| | 2x8 | 32K | 512K | 106.37 | **1.38x** | 102.34 | **1.43x** | 107.76 | 2.01x |

Table 2: The maximal sequence length Per GPU supported by LIGHTSEQ and Megatron-LM with tensor parallelism and pipeline parallelism on 16xA100 40GB GPUs. LIGHTSEQ supports 512K sequence length in all models, while Megatron-LM strategy maximal sequence length decreases with fewer heads, with either data parallelism or pipeline parallelism.

| | Llama-16H | Llama-8H | Llama-4H | Llama-2H |
|---|---|---|---|---|
| Megatron TP+DP | 512K | 256K | 128K | 64K |
| Megatron-LM TP+PP | 512K | 256K | 256K | 128K |
| LIGHTSEQ | 512K | 512K | 512K | 512K |

**Grouped-query attention (GQA).** On LLama-GQA model, our method achieves better speedup because our communication of key and value vectors significantly reduces. Note that our communication time is proportional to the sum of query, key, value, and output (for load balancing) vectors, where reducing key and value sizes to 8 almost half-en our communication time. On the contrary, the communication time in Megatron-LM does not decrease because its communication happens outside of the attention module, i.e. not influenced by optimization inside the attention module. Thus, its overall training run-time does not decrease as much as LIGHTSEQ.

We take the 4K per-GPU sequence length and 2x8 GPUs as an example for analysis. In the MHA experiment, the communication in a forward and a backward pass of a single attention module is roughly 143ms and the computation time is roughly 53ms. In addition, our overlapping technique is able to hide 45ms into the computation, resulting in a total run-time of 151ms and a net communication overhead of 98 ms. As a reference, the communication in Megatron-LM takes 33ms, which is why Megatron-LM is faster than LIGHTSEQ under this particular setting in the MHA experiment. When considering the GQA case, the communication in LIGHTSEQ roughly reduces to 71 ms. Overlapping with the computation, the communication overhead is now less than that of Megatron-LM. Combined with the checkpointing technique, we are seeing a positive speedup gain at 4K per-GPU sequence length. As the sequence length increases, our overlapping technique, driven by the fact that computation time surpasses communication time, and our checkpointing method, due to the rising ratio of a single attention forward, both contribute to greater speedup. Overall, we can observe speedups up to **1.52**× on the cross-node setting, making an additional eight percent enhancement compared to the results in the MHA experiment of the same setting.

Figure 5: Ablation on the effect of balanced schedule (left) and the effect of overlapping (right).

**In support of arbitrary numbers of heads.** With Llama-33H models, Megatron-LM exhibits an additional performance decline compared to LIGHTSEQ. This is due to its requirement to pad the number of attention heads so that the number of attention heads is divisible by the number of devices. On the other hand, LIGHTSEQ does not need to partition attention heads and can support an arbitrary number of heads efficiently. For instance, when using 8 GPUs, Megatron-LM must pad the attention heads to 40, resulting in 21.2% of the computation being wasted. In the case of 16 GPUs, Megatron-LM is compelled to pad the attention heads to 48, leading to a more substantial computation wastage of 45.5%. This roughly corresponds to a 1.21× or 1.45× increase in run-time compared to LIGHTSEQ when training a Llama-7B model. This performance degradation of Megatron-LM is primarily because the training time is dominated by the attention module's computation time when scaling to longer sequence lengths. Empirically, we observe a **1.50×** and **2.01×** speedup (an additional 20% and 45% speedup compared to Llama-7B cases, aligned with the theoretical analysis).

### 4.2 SCALING BEYOND THE NUMBER OF HEADS.

Assuming the number of heads being a multiple of the tensor parallelism degree constraints Megatron-LM to scale its tensor parallelism degree beyond the number of heads, thus limiting its scaling ability to longer sequence lengths. When the number of GPUs exceeds the number of attention heads, there will be three possible solutions to use Megatron-LM. First, the user can pad dummy heads as in the Llama-33H scenario. However, when scaling to longer sequences, the percentage of dummy heads padded almost directly translates to the percentage of slowdown. For instance, for Llama-8H, this solution pads 2× dummy heads and would almost translate to a 2× slowdown, which is very inefficient. Second, the user can use data parallelism for excess GPUs. For instance, a user with 16 GPUs can choose to use 4-way data parallelism and 4-way tensor parallelism on the Llama-4H model. Since data parallelism does not partition the activation, the system can only support sequences as if the user only has 4 GPUs. Lastly, the user may choose to use pipeline parallelism to partition activation. However, the memory usage at each stage of the pipeline is not evenly distributed, still limiting the maximal sequence length supported. In particular, the first pipeline stage usually stores more activations because it will hold the most active micro-batches. For instance, in the Llama-2H experiment, we find that different stages consume from 18GB to 32GB in a 64K sequence length (Section E). In addition, using pipeline parallelism introduces an extra fraction of GPU idle time. We demonstrate the effect of using the latter two solutions in Table 6. In 16 A100 40GB GPUs, LIGHTSEQ supports the training of 2× and 8× longer sequences.

### 4.3 ABLATION STUDY

**Effect of load balancing.** We study the effect of load balancing using the forward pass of an attention operation in Llama-7B model, on 8 A100 40GB GPUs. The backward pass follows a similar analysis. With an unbalanced schedule (Figure 1), the total work done is 36, where the total work could be done in 8 units of time is 64. Thus, the expected maximal speedup is 4.5x. In the balanced schedule, the expected maximal speedup is 7.2x. We scale the total sequence length from 4K to 256K. The unbalanced version saturates in 4.5x speedup compared to a single GPU implementation, while the balanced version saturates 7.5x [2] speedup. Both of them align with our earlier theoretical analysis and show the importance of our balanced scheduling.

**Effect of overlapping communication and computation.** We study the benefits of overlapping communication on Llama-7B and 2 DGX boxes. We find that overlapping greatly reduce the communication overhead. For instance, on a global sequence length of 128K, the communication overhead is reduced from 105% to 44%. This overlapping scheme maximizes its functionality when the

---

[2] We find the single machine attention flops drop with very long sequence length, resulting in a slightly higher speedup than assuming its perfect scalability.

communication overhead is less than 100%, where all communication can be potentially overlapped. Empirically, we find the system only exhibits 8% and 1% overhead in these cases, showing a close performance to an ideal system without communication.

**Effect of materialization-aware checkpointing.** We show in Table. 3 the ablation results of our rematerialization-aware gradient checkpointing. Our method achieves 1.16x, 1.24x, and 1.31x speedup at the sequence length of 8K, 16K, and 32K per GPU respectively. The materialization-aware checkpointing strategy speeds up more at longer sequence lengths because it saves an entire attention forward which dominates the computation at longer sequence lengths.

Table 3: Ablation study on the effect of the rematerialization-aware gradient checkpointing on 8 A100s in a single node with a batch size of 1. We report the end-to-end run time in seconds and show the speedup of our gradient checkpointing strategy ("Our ckpt") over the HuggingFace gradient checkpointing strategy ("HF ckpt").

| Ckpt Method | Sequence Length Per GPU | | | | | |
|:---:|:---:|:---:|:---:|:---:|:---:|:---:|
| | 1K | 2K | 4K | 8K | 16K | 32K |
| HF ckpt | 0.84 | 1.29 | 2.64 | 6.93 | 21.44 | 76.38 |
| Our ckpt | 0.84 | 1.36 | 2.50 | 5.98 | 17.26 | 58.46 |
| Speedup | 1.0x | 0.94x | **1.06x** | **1.16x** | **1.24x** | **1.31x** |

## 4.4 COMPARISON WITH DEEPSPEED UYLESS

DeepSpeed-Ulysses [3] is a concurrent open-sourced implementation, which uses all-to-all communication primitive to reduce the communication volume. In our testing, we verified that their communication is lower than Megatron-LM. Yet, as it is also partitioning the attention head dimension, it suffers from similar problems as analyzed above. We provide some end-to-end comparisons in Appendix B. We note that the communication in DeepSpeed Ulysses can be faster than LIGHTSEQ, especially with shorter context length and slower network, where the overlapping technique in LIGHTSEQ cannot perfectly hide all the communication. This can be potentially addressed by optimizing the P2P communication as discussed above.

## 4.5 DISCUSSION

In this section, we discuss the future directions that can further improve LIGHTSEQ.

**Optimizing P2P communication and better support for shorter context length and lower bandwidth region.** As shown in §4.1, LIGHTSEQ may be slower in shorter context length and MHA models (Llama-7B on per GPU sequence length 4K). Based on our preliminary investigation, this is because our usage of P2P is not as optimized as primitives used in tensor model parallelism, such as all-gather kernels .For instance, they are not aware of the underlying cluster topology. For the same reason, we also observed that the current P2P optimization has not achieved the theoretical $4.7\times$ speedup in a very low bandwidth region (e.g. in our in-house cluster). In the future, we plan to implement the P2P scheduling in a topology-aware way to further improve the communication time.

## 5 CONCLUSION

In this work, we introduce LIGHTSEQ, a sequence parallel prototype for long-context transformer training. LIGHTSEQ presents novel system optimizations including load balancing for causal language modelings, overlapped communication with computation in the distributed attention computation, and a re-materialization-aware checkpointing strategy. Our experiments evaluate multiple families of transformer models and on different cluster types, showing that it achieves up to $2.01\times$ speedup and scales up to 8x longer sequences, compared to another popular system, Megatron-LM,. Future directions include implementing topology-aware P2P operations to further reduce training time in lower sequence lengths.

---

[3]`https://github.com/microsoft/DeepSpeed/tree/master/blogs/deepspeed-ulysses`

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

## Appendix

## A    Using DistAttn in LightSeq

---

**Algorithm 1** DistAttn in LightSeq (forward pass)

---

**Require:** Matrices $\mathbf{Q}^p, \mathbf{K}^p, \mathbf{V}^p \in \mathbb{R}^{\frac{N}{\mathbb{P}} \times d}$ in HBM, block sizes $B_c$, $B_r$, rank
1: **function** STANDALONE_FWD(q, k, v, o, $\ell$, m, causal, last)
2:     Divide $q$ into $T_r = \left\lceil \frac{N}{\mathbb{P} B_r} \right\rceil$ blocks $q_1, \ldots, q_{T_r}$ of size $B_r \times d$ each,
3:     and divide $k, v$ in to $T_c = \left\lceil \frac{N}{\mathbb{P} B_c} \right\rceil$ blocks $k_1, \ldots, k_{T_c}$ and $v_1, \ldots, v_{T_c}$, of size $B_c \times d$ each.
4:     Divide the output $o \in \mathbb{R}^{\frac{N}{\mathbb{P}} \times d}$ into $T_r$ blocks $o_i, \ldots, o_{T_r}$ of size $B_r \times d$ each, and divide the logsumexp $L$ into $T_r$ blocks $L_i, \ldots, L_{T_r}$ of size $B_r$ each.
5:     **for** $1 \leq i \leq T_r$ **do**
6:         Load $q_i$ from HBM to on-chip SRAM.
7:         Load $o_i \in \mathbb{R}^{B_r \times d}, \ell_i \in \mathbb{R}^{B_r}, m_i \in \mathbb{R}^{B_r}$ from HBM to on-chip SRAM as $o_i^{(0)}, \ell_i^{(0)}, m_i^{(0)}$.
8:         **for** $1 \leq j \leq T_c$ **do**
9:             **if** causal and $i \leq j$ **then**
10:                 Continue
11:             **end if**
12:             Load $k_j, v_j$ from HBM to on-chip SRAM.
13:             On chip, compute $s_i^{(j)} = q_i k_j^T \in \mathbb{R}^{B_r \times B_c}$.
14:             On chip, compute $m_i^{(j)} = \max(m_i^{(j-1)}, \text{rowmax}(s_i^{(j)})) \in \mathbb{R}^{B_r}$, $\tilde{p}_i^{(j)} = \exp(S_i^{(j)} - m_i^{(j)}) \in \mathbb{R}^{B_r \times B_c}$ (pointwise), $\ell_i^{(j)} = e^{m_i^{j-1} - m_i^{(j)}} \ell_i^{(j-1)} + \text{rowsum}(\tilde{p}_i^{(j)}) \in \mathbb{R}^{B_r}$.
15:             On chip, compute $o_i^{(j)} = \text{diag}(e^{m_i^{(j-1)} - m_i^{(j)}})^{-1} o_i^{(j-1)} + \tilde{p}_i^{(j)} v_j^p$.
16:         **end for**
17:         On chip, compute $o_i = \text{diag}(\ell_i^{(T_c)})^{-1} o_i^{(T_c)}$.
18:         Write $o_i$ to HBM as the $i$-th block of $o$.
19:         **if** last **then**
20:             On chip, compute $L_i = m_i^{(T_c)} + \log(\ell_i^{(T_c)})$.
21:             Write $L_i$ to HBM as the $i$-th block of $L$.
22:         **end if**
23:     **end for**
24:     Return $o, \ell, m$ and the logsumexp $L$.
25: **end function**
26: Initialize $\mathbf{O}^p = (0)_{\frac{N}{\mathbb{P}} \times d} \in \mathbb{R}^{\frac{N}{\mathbb{P}} \times d}, \ell^{(p)} = (0)_{\frac{N}{\mathbb{P}}} \in \mathbb{R}^{\frac{N}{\mathbb{P}}}, m^p = (-\infty)_{\frac{N}{\mathbb{P}}} \in \mathbb{R}^{\frac{N}{\mathbb{P}}}$.
27: $\mathbf{O}^p, \ell^p, m^p, L^p = \text{standalone\_fwd}(\mathbf{Q}^p, \mathbf{K}^p, \mathbf{V}^p, \mathbf{O}^p, \ell^p, m^p, \text{True, p=1})$
28: **for** $1 \leq r < p$ **do**
29:     Receive $\mathbf{K}^r$ and $\mathbf{V}^r$ from **Remote** worker $r$ into HBM.
30:     $\mathbf{O}^p, \ell^p, m^p, L^p = \text{standalone\_fwd}(\mathbf{Q}^p, \mathbf{K}^y, \mathbf{V}^y, \mathbf{O}^p, \ell^p, m^p, \text{False, r=(p-1)})$
31:     Delete $\mathbf{K}^r$ and $\mathbf{V}^r$ from HBM.
32: **end for**
33: Return the output $\mathbf{O}^p$ and the logsumexp $L$.

---

In this section, we provide more details of DistAttn, and how it can be used with the outer LightSeq logic of the forward pass (Alg 1). For conceptual simplicity, we demonstrate it in the most vanilla version, without the actual scheduling (e.g. load balancing and overlapping). We also demonstrate it with the causal language modeling objective. The standalone attention is mainly borrowed from the FlashAttention2 paper (Dao, 2023). To make it compatible with DistAttn, we mainly revised the several points:

1. Accumulate results statistics $o$, $m$ and $l$ from previous computation, instead of initializing them inside the function.

2. Pass an extra argument "last", which means whether this is the last chunk of attention computation. Only when it is true, we compute the logsumexp $L$.

At a high level, on a worker $p$, LIGHTSEQ first initializes local statistics $m, l, L$. Then LIGHTSEQ loops over all its previous workers. In each iteration, it fetches the key and the value from a worker and invokes the revised standalone attention to update local statistics. At the end of the iteration, it needs to delete the remote key and value from HBM so that the memory does not accumulate. At the last iteration of the loop, it additionally calculates the logsumexp according to the final $m$ and $l$ (the "last" variable in the algorithm). At the end of the forward pass, worker $p$ has the correct $m, l, L$. The backward pass is similar and conceptually simpler because we do not need to keep track of statistics such as $m$ and $l$. Instead, we only need to use the logsumexp stored in the forward pass.

## B    COMPARISON WITH DEEPSPEED ULYSSES

We run a subset of the experiments compared with DeepSpeed-Ulysses. Firstly, DeepSpeed-Ulysses does reduce the communication overhead, and thus better than Megatron-LM on scenarios listed in Table 4. LIGHTSEQ achieves better performance than DeepSpeed-Ulysses on longer sequences or models with a more general number of heads (e.g. Llama-33H). We also note that DeepSpeed-Ulysses can not scale beyond the number of attention heads because it also relies on sharding the attention heads. However, we need to point out that in shorter sequences and MHA models (where LIGHTSEQ does not have a communication advantage, compared to GQA/MQA models), the communication primitives used in DeepSpeed-Ulysses are more advantageous. We leave our further optimization in P2P in shorter sequences and MHA models as an exciting future work.

## C    LOAD-BALANCING ALGORITHM FOR CAUSAL MODELING

In this section, we detail the design of our load-balancing algorithm for causal modeling. We show the workload of each worker in all time steps in Figure 6 (before applying load-balancing) and Figure 7 (after applying load-balancing) in an 8-worker scenario. The communication schema is also reflected in both figures by comparing the tensors each worker holds at the consecutive two time steps.

## D    COMPARISON WITH RING SELF-ATTENTION (RSA)

Ring self-attention is among the first sequence parallelism work for Transformers, proposed in Li et al. (2021). It communicates tensors in a ring fashion. Firstly, we report the maximal sequence length of RSA and LIGHTSEQ in Table 5, and found that LIGHTSEQ supports at least 8x longer sequences than RSA. This is mainly because RSA is not natively compatible with memory-efficient attention (i.e. it modifies the attention computation). We further measure the iteration time with the maximal sequence length RSA can support in Table 6, and found that LIGHTSEQ is 4.45x - 5.64x faster than RSA. This is mainly because LIGHTSEQ has optimized sequence parallelism in (1) causal language objective (  2x speedup), and (2) memory-efficient attention (Dao, 2023). In explanation, memory-efficient attention also speeds up the attention computation by carefully managing the IO during attention, as pointed out in (Dao et al., 2022; Dao, 2023).

## E    MEMORY CONSUMPTION FOR PIPELINE PARALLELISM

In this section, we show the memory consumption of Megatron-LM when training with tensor parallelism and pipeline parallelism. As presented in table 7, memory consumption are uneven across different pipeline stages, making scaling through pipeline parallelism hard.

| Method | # GPUs | Sequence Length | | Time | Speedup |
| --- | --- | --- | --- | --- | --- |
| | | Per GPU | Total | | |
| Llama-7B | | | | | |
| Megatron-LM | 2x8 | 4K | 64K | 5.29 | 1.0x |
| | 2x8 | 8K | 128K | 14.26 | 1.0x |
| | 2x8 | 16K | 256K | 43.44 | 1.0x |
| | 2x8 | 32K | 512K | 147.06 | 1.0x |
| DeepSpeed-Ulysses | 2x8 | 4K | 64K | 4.29 | **1.23x** |
| | 2x8 | 8K | 128K | 11.61 | **1.23x** |
| | 2x8 | 16K | 256K | 37.53 | 1.16x |
| | 2x8 | 32K | 512K | 134.09 | 1.10x |
| LIGHTSEQ | 2x8 | 4K | 64K | 6.85 | 0.77x |
| | 2x8 | 8K | 128K | 12.75 | 1.12x |
| | 2x8 | 16K | 256K | 30.21 | **1.44x** |
| | 2x8 | 32K | 512K | 106.37 | **1.38x** |
| Llama-33H | | | | | |
| Megatron-LM | 2x8 | 4K | 64K | 7.52 | 1.0x |
| | 2x8 | 8K | 128K | 20.63 | 1.0x |
| | 2x8 | 16K | 256K | 62.78 | 1.0x |
| | 2x8 | 32K | 512K | 216.70 | 1.0x |
| DeepSpeed-Ulysses | 2x8 | 4K | 64K | 6.42 | **1.17x** |
| | 2x8 | 8K | 128K | 17.47 | 1.18x |
| | 2x8 | 16K | 256K | 56.63 | 1.11x |
| | 2x8 | 32K | 512K | 202.89 | 1.07x |
| LIGHTSEQ | 2x8 | 4K | 64K | 7.03 | 1.07x |
| | 2x8 | 8K | 128K | 13.12 | **1.57x** |
| | 2x8 | 16K | 256K | 31.33 | **2.00x** |
| | 2x8 | 32K | 512K | 107.76 | **2.01x** |

Table 4: Per iteration wall-clock time of LIGHTSEQ, Megatron-LM (Korthikanti et al., 2023) and DeepSpeed Ulysses (Unit: seconds). Speedup in bold denotes the better of the three systems. We calculate the speedup based on Megatron-LM iteration time. Time measured with cluster 2, 2 DGX boxes.

Table 5: Maximal sequence length on Llama-7B on the DGX (A100-80GB) cluster.

| | 1 Node (8 GPUs) | 2 Nodes (16 GPUs) |
| --- | --- | --- |
| RSA | 32K | 64K |
| LIGHTSEQ | > 256K | > 512K |

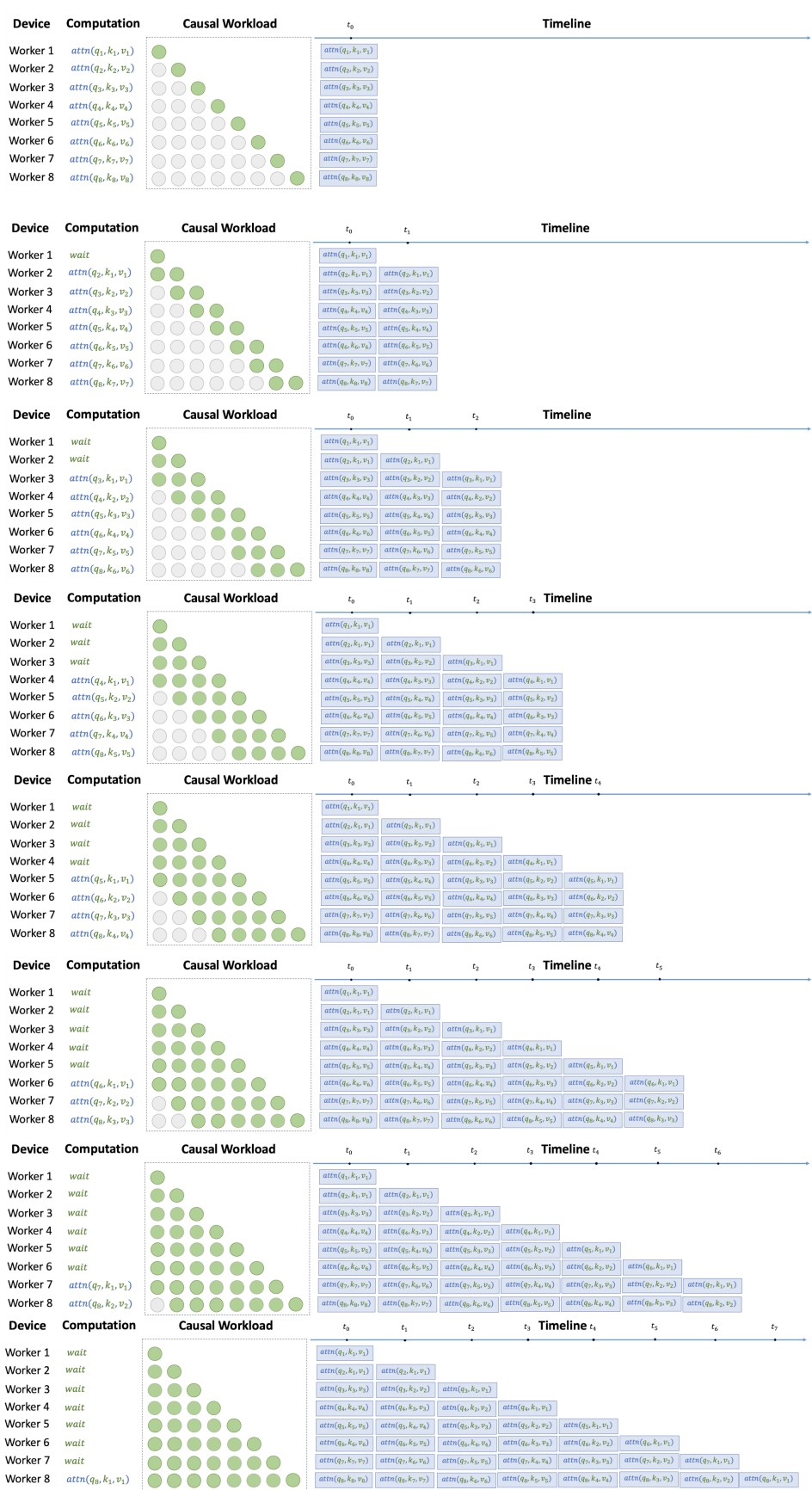

Figure 6: Illustration of DISTATTN before applying load-balancing on 8 workers.

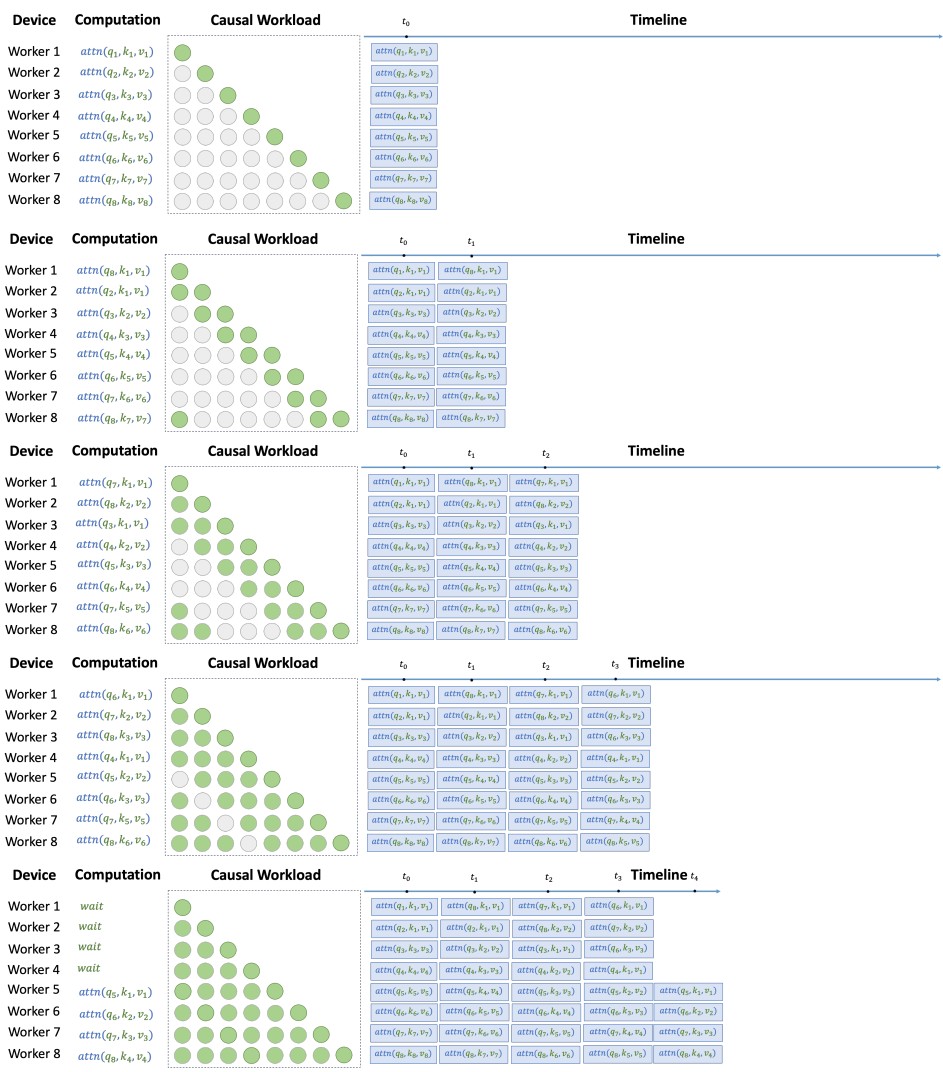

Figure 7: Illustration of DISTATTN after applying load-balancing on 8 workers.

Table 6: Per iteration time comparison with RSA (seconds) on the DGX clusters.

|  | 1 Node (32K) | 2 Nodes (64K) |
|---|---|---|
| RSA | 14.10 | 30.49 |
| LIGHTSEQ | 2.50 | 6.85 |
| Speedup | 5.64x | 4.45x |

Table 7: The memory consumption of Megatron-LM when training Llama-2H with tensor parallelism (degree=2) and pipeline parallelism (degree=8) on 16xA100 40GB GPUs at the sequence length of 128K. The memory consumption is highly uneven across pipeline stages.

|  | Worker 1 | Worker 2 | Worker 3 | Worker 4 | Worker 5 | Worker 6 | Worker 7 | Worker 8 |
|---|---|---|---|---|---|---|---|---|
| node 1 | 31.5GB | 31.4GB | 28.7GB | 28.7GB | 26.0GB | 26.0GB | 24.6GB | 24.6GB |
| node 2 | 21.8GB | 21.8GB | 20.5GB | 20.5GB | 17.9GB | 17.8GB | 32.0GB | 32.1GB |

