# OpenReview forum: "LightSeq: Sequence Level Parallelism for Distributed Training of Long Context Transformers"
_ICLR.cc/2024/Conference — Submitted to ICLR 2024_

### Official Review · Reviewer_xpDg · 2023-11-01

**Soundness:** 3 good
**Presentation:** 3 good
**Contribution:** 2 fair
**Rating:** 6
**Confidence:** 5

**Summary:**

The authors introduce LightSeq, a distributed training approach for long-context LLMs.
LightSeq uses sequence parallelism over multiple GPUs, i.e. splitting query, key and value along the sequence dimension. Their approach DistAttn then computes the attention matrix chunk wise.
Their approach improve performance for large context lengths compared to parallelization over attention heads such as Megatron-LM as the degree of parallelization is not limited by the number of heads.
By overlapping communication and computation, improving load-balancing for unidirectional models, and adjusting checkpointing for flash-attention, LightSeq achieves a speedup between 0.77 and 2.01 over Megatron-LM when training Llama-7B based models.

**Strengths:**

- The research topic is of importance and the presented approach appears useful. Parallelization is essential in training LLMs and parallelization over the sequence length can be important with the ever-increasing context lengths, as is also demonstrated by the experiments in this paper.
- The paper is well written and easy to follow. It clearly communicates the motivation and contributions and gives an easy-to-understand explanation of the introduced approaches.
- The approach seems overall well-grounded and optimized, employing several techniques to increase performance and reduce idle time.
- The experimental results demonstrate a significant speedup over Megatron-LM, a widely used parallelization strategy for transformer training.
- The authors provide source-code, thereby supporting reproducibility of their approach

**Weaknesses:**

One issue is the assumption of causal learning tasks and thus the limitation of the approach to uni-directional Transformer architectures. The approach does not translate to bi-directional models, such as BERT, where long sequences are becoming a problem just as well.

My main concern however is the limited experimental evaluation, which focusses solely on comparing to Megatron-LM and only on Llama model variants, even though the authors mention other sequence-parallel approaches .
For one giving only the speedup over a single other parallel approach (Table 1) limits the general comparability:
- For one including the speedup over sequential BP would be helpful to judge how well both of these approaches scale.
- A comparison to Li et al., which the authors mention, would be highly necessary. The authors state that this approach is not optimized for unidirectional, long-context training, so the benefit of LightSeq over this approach for models such as Llama would be valuable.
- And while the authors argue that pipeline parallelism is limited by its unbalanced memory consumption, it would be good to back this up with experimental results. A comparison to TeraPipe [1], which employs token-level pipelining, would also be interesting.

Furthermore, the evaluated model architectures seem highly tailored to the specific aspects of LightSeq, which again does not demonstrate any generalization.
- The 33H case seems rather constructed and unlikely in practice (why would you chose such a number specifically?). It is a valid limitation of parallelization over the heads but should not be that big a problem in actual use, reducing the realistic speedup to ~1.5x (which is still substantial). Also, the title of the paragraph does not really fit the content as it never gives any support why one would want to use an arbitrary numbers of heads.
- In that regard the suggestion of "scaling" Megatron by adding dummy heads seems unnecessary, the authors even discuss how inefficient this would be. I don't believe it is necessary to discuss this first "option" at all.


Further issues
- Quite a lot of detail on the algorithmic implementation of DistAttention, e.g. the communication scheme, is lacking. One has to dig through the source code to get a glimps of what is going on
- In contrast to the other sections, section 4 seems unpolished and is often hard to understand
- no links between different sections, some Figures are never referred to, inconsistent punctuation when referring to figures and tables. This could all be solved by using a package like cleverref or autoref.
- the difference between the different models in the model setup is not described clearly enough, GQA should be cited and explained a bit better, the 33H model is only clear from context, and it is not clear how exactly the 16H-2H models are scaled "properly"
- The example analysis comparing MHA and GQA in Section 4.1 is a bit hard to follow and it would probably help to give the actual runtimes and a table or figure.
- in Section 4.3 when discussing the effect of load balancing, a diagram illustrating the work assigned to each time step would help visualizing the difference between the schedules. It would also help to remind the reader again, that the balanced schedule completes the 36 work units in 5 time steps, thus resulting in a maximum speedup of 7.2, instead of having the reader backtrack to understand where the 7.2 came from.
- Section 4.4 seems a bit odd, it is titled "Discussion" but contains a mix of future work, introducing a very brief comparison to DeepSpeed Ulysses which is never mentioned before or after, and reiterating on the drawbacks of pipeline parallelism without given much new information that had not already been stated in Section 4.2.

**Questions:**

- Your approach for DistAttn seems to me very similar to what FlashAttention2 does to parallelise on a single GPU. Can you please elaborate a bit more on the algorithmic differences between the two?
- LightSeq is especially advantageous when the sequences are long and there are few heads or the number of heads is not well divisible by the number of GPUs. Would it be possible/make sense to use tensor-parallelism (in addition to the sequence-parallelism) in the DistAttn to retain the advantages of Megatron-LM and also scale well in scenarios with shorter sequences and many heads?

---

> ### Author Response · Authors · 2023-11-20
> **Reply to Reviewer xpDg (Part 1)**
>
> We thank the reviewer for the insightful comments! We would like to answer your questions and address your concerns in this response:
>
> **Q1:** Your approach for DistAttn seems to me very similar to what FlashAttention2 does to parallelize on a single GPU. Can you please elaborate a bit more on the algorithmic differences between the two?
>
> **A1:** In the distributed setting, unbalanced computation and the cross-GPU communication overhead become two challenges that do not exist in the single-GPU case. Therefore, we propose and implement a load-balancing scheduling and an asynchronous communication schema to overcome this. DistAttn without these two optimizations can be understood as a distributed version of FlashAttention2. We extended the FlashAttention2 kernel to support multi-GPU computation by sharding the outer loop onto different devices and communicating the keys and values upon need.
>
> **Q2:** LightSeq is especially advantageous when the sequences are long and there are few heads or the number of heads is not well divisible by the number of GPUs. Would it be possible/make sense to use tensor-parallelism (in addition to the sequence-parallelism) in the DistAttn to retain the advantages of Megatron-LM and also scale well in scenarios with shorter sequences and many heads?
>
> **A2:** This is a great suggestion that we do recommend in practice to replace (instead of combining at the same time) TP with DistAttn at long sequences as DistAttn requires less communication and supports scaling beyond the number of heads.
>
> **Q3:** The 33H case seems rather constructed and unlikely in practice (why would you choose such a number specifically?). It is a valid limitation of parallelization over the heads but should not be that big a problem in actual use, reducing the realistic speedup to ~1.5x (which is still substantial)
>
> **A3:** Yes, we designed the 33-head model to illustrate the advantages of keeping the distributed training design agnostic to the model configuration. We picked the number 33 for this illustration which may sound unusual. However, we would like to note that it is a common case in practice to have a non-power-of-2 number of heads as we discussed in the second paragraph of the introduction. For example, GPT-2-XL has 25 attention heads, GPT-2 has 12 attention heads, Llama-33B and its fine-tuned versions (e.g., Tulu-30B) have 52 attention heads, Whisper-large has 20 attention heads, and Falcon-7B has 71 attention heads. We would also note that our design can facilitate future model architectures as we discussed in the second paragraph of the introduction:
> > Moreover, many works have shown that the future Transformer architecture design may have even fewer attention heads. For example, Bian et al. (2021) demonstrate that Transformers with a single head outperforms its multi-head counterparts, representing a challenging scenario for solutions like Megatron-LM.
>
> **Q4:** One issue is the assumption of causal learning tasks and thus the limitation of the approach to uni-directional Transformer architectures. The approach does not translate to bi-directional models, such as BERT, where long sequences are becoming a problem just as well.
>
> **A4**: We thank the reviewer for pointing this out. We focus on the optimization for the GPT-based model (specifically, the Llama models) as it’s the most popular model architecture in recent days. The evaluation of bi-directional models is out-of-scope in this paper. However, we respectfully disagree that our approach is therefore only limited to uni-directional Transformer architectures. The DistAttn can be easily applied to bi-directional models by simply disabling the load-balancing scheduler. The load-balancing optimization is proposed to address the bubble caused by causal modeling. But there’s no bubble when handling bi-directional attention, in which case the vanilla design of DistAttn w/o load-balancing scheduling will be good enough.

---

> > ### Author Response · Authors · 2023-11-20
> > **Reply to Reviewer xpDg (Part 2)**
> >
> > **Q5:** My main concern however is the limited experimental evaluation, which focuses solely on comparing to Megatron-LM and only on Llama model variants, even though the authors mention other sequence-parallel approaches.
> >
> > **A5:** We also compared with DeepSpeed Ulysses (one concurrent work within one month of the submission deadline) in Appendix B Table 4, we moved the discussion and comparison with DeepSpeed Ulysses to the experiment section in the updated version. We show LightSeq has up to 1.44x speedup in the intra-node setting and up to 2.01x speedup in the cross-node setting.
> >
> > In addition, we added a comparison with the 4D sequence parallelism work over Llama-7B, and quantitatively shows that LightSeq supports at least 8x longer sequences and roughly 5x faster training.
> >
> > We chose the Llama model as the model structure because of its popularity, and evaluate several variants of Llama that have different system characteristics. However, we point out that our design is for all Transformer models that are usually composed of attention modules, feed-forward modules, layer normal, etc. The key difference between the Llama models and the others (e.g., the GPT models) is the feed-forward module which we show isn’t the bottleneck of long-context training and we can reasonably expect similar performance when shifting the model architecture from Llama to others.
> >
> > ### Table 5: Maximal sequence length on Llama-7B on the DGX (A100-80GB) cluster.
> >
> > |                | 1 Node (8 GPUs) | 2 Nodes (16 GPUs) |
> > |----------------|-----------------|-------------------|
> > | RSA            | 32K             | 64K               |
> > | LIGHTSEQ       | > 256K          | > 512K            |
> >
> > ### Table 6: Per iteration time comparison with RSA (seconds) on the DGX clusters.
> >
> > |                | 1 Node (32K) | 2 Nodes (64K) |
> > |----------------|--------------|---------------|
> > | RSA            | 14.10        | 30.49         |
> > | LIGHTSEQ       | 2.50         | 6.85          |
> > | Speedup        | 5.64x        | 4.45x         |
> >
> > **Q6:** While the authors argue that pipeline parallelism is limited by its unbalanced memory consumption, it would be good to back this up with experimental results.
> >
> > **A6:** We added a discussion of the unbalanced memory consumption in the related work section.
> > Pipeline parallelism keeps a high memory pressure to the first stage when applying interleaved pipeline parallelism to minimize the computation bubble. We back this up by experiments in Table 7 in the updated paper. When disabling the interleaved pipeline scheduling, the computation bubble increases, causing inefficient computation. We refer to Korthikanti et al.[1] for more details.
> >
> > ### Table 7: The memory consumption of Megatron-LM when training Llama-2H with tensor parallelism (degree=2) and pipeline parallelism (degree=8) on 16xA100 40GB GPUs at the sequence length of 128K. The memory consumption is highly uneven across pipeline stages.
> >
> > |        | Worker 1 | Worker 2 | Worker 3 | Worker 4 | Worker 5 | Worker 6 | Worker 7 | Worker 8 |
> > |--------|----------|----------|----------|----------|----------|----------|----------|----------|
> > | node 1 | 31.5GB   | 31.4GB   | 28.7GB   | 28.7GB   | 26.0GB   | 26.0GB   | 24.6GB   | 24.6GB   |
> > | node 2 | 21.8GB   | 21.8GB   | 20.5GB   | 20.5GB   | 17.9GB   | 17.8GB   | 32.0GB   | 32.1GB   |
> >
> > [1] Korthikanti, Vijay Anand, et al. "Reducing activation recomputation in large transformer models." Proceedings of Machine Learning and Systems 5 (2023).
> >
> > **Q7:** Quite a lot of detail on the algorithmic implementation of DistAttention, e.g. the communication scheme, is lacking.
> >
> > **A7:**  We thank the reviewer for pointing this out. We added detailed explanations of the computation and communication scheme in Appendix C and updated Figure 7 for clearer illustration.
> >
> > **Q8:** The difference between the different models in the model setup is not described clearly enough, GQA should be cited and explained a bit better, the 33H model is only clear from context, and it is not clear how exactly the 16H-2H models are scaled "properly"
> >
> > **A8:** We have included more explanation about the GQA (and cited) and Llama-33H in the experiment setup section in the updated version. For 16H-2H models, we keep the number of attention heads the same in the entire model. For instance, Llama-7B has 32 attention heads per layer and 32 layers, Llama-16H has 16 attention heads per layer and thus 64 layers in total, according to [1]. We have also updated this in the experiment section.
> >
> > [1] Multi-head or single-head? an empirical comparison for transformer training.

---

> > > ### Author Response · Authors · 2023-11-20
> > > **Reply to Reviewer xpDg (Part 3)**
> > >
> > > **Q9:** In Section 4.3 when discussing the effect of load balancing, a diagram illustrating the work assigned to each time step would help visualize the difference between the schedules.
> > >
> > > **A9:** We updated the illustration in Appendix C to also contain time-step information, making it easier for readers to understand! Specifically, Figure 6 shows DistAttn design before applying load-balancing and Figure 7 shows DistAttn after applying load-balancing. The communication scheduling is also reflected in both figures by comparing the tensors each workder holds at two consecutive time steps.
> > >
> > > **Q10:** Section 4.4 seems a bit odd, it is titled "Discussion" but contains a mix of future work, introducing a very brief comparison to DeepSpeed Ulysses which is never mentioned before or after, and reiterating the drawbacks of pipeline parallelism without giving much new information that had not already been stated in Section 4.2.
> > >
> > > **A10:** Thanks for pointing this out! We apologize for the confusion. We have reorganized the paper to make the content fit the flow better. Specifically, we moved the discussion of comparison with DeepSpeed-Ulysses to the experiment section and the discussion of the pipeline parallelism to the related work section, which fits better the paper flow.

---

> ### Author Response · Authors · 2023-11-21
> **Thank you and let us know any further comments**
>
> Dear reviewer xpDg,
>
> We are really encouraged by your helpful and positive feedback, and sincerely appreciate your efforts. We hope our answer, updated experiments and the corresponding manuscript makes the paper clearly and stronger. Please let us know if you have any further comments, and we are more than happy to address them.
>
> Best,
>
> The authors

---

> > ### Author Response · Authors · 2023-11-22
> > **We kindly request the reviewer to participate in the discussion before the deadline.**
> >
> > Dear reviewer,
> >
> > We thank again for your previous time and insightful comments. As the deadline of the rebuttal period approaching, we appreciate your feedback on our response, and we're more than happy to address any other last-minute question.
> >
> > Happy Thanksgiving week,
> >
> > The authors

---

### Official Review · Reviewer_AsB4 · 2023-11-02

**Soundness:** 3 good
**Presentation:** 3 good
**Contribution:** 2 fair
**Rating:** 3
**Confidence:** 4

**Summary:**

This work, LIGHTSEQ, proposes a sequence parallelism prototype for long-context transformer
training. Three key elements are introduced: 1) distributed attention with load balancing for causal language
modelings, 2) distributed attention with overlapped communication with computation, and 3) a re-materialization-aware checkpointing strategy. This work's experiments on 16 A100s show decent speedup over Megatron-LM on Llama models as well as enabled longer sequences.

**Strengths:**

+. Proposed a new loading balancing schedule dedicated to distributed attention by shifting busy workers' q/k/v compute to idle workers.

+. Developed a system optimization of overlapping remote q/k/v with local compute by using two CUDA streams.

+. Improved huggingface's gradient checkpointing by leveraging FlashAttention's recompute feature in backward compute.

+. Has open-source code implementation

+. Evaluation on real clusters

**Weaknesses:**

-. **Overstatement**:

> "we propose partitioning solely the input tokens (i.e., sequence parallelism)"

why sequence parallelism is proposed in this work, instead of enhanced by this work?

> "We present a solution that is agnostic to the model architecture"

why sequence parallelism by default is NOT agnostic to model architecture (i.e., like data parallel, just replicate the layer across devices)?

> Figure 4

By default, FlashAttention uses recompute during backward pass. How does this become a contribution of this work?

> Figure 1 (left)

By default, using sequence parallelism for the attention layer should be like this: replicate FFN and projection, and communication (k, v) for attention op. why did this scheme become a proposal for this work?

-. **Unclear load-balanced scheduling**:

> Figure 1 (right)

Besides shifting worker 6,7,8's compute on (kv1, kv2, kv3) to worker 1,2,3, can we also shift worker 6,7,8's compute on (kv6, kv7, kv8)?

Is there a way to optimally find the best compute workload to shift?

How does the DistAttn & load balancing work for attention op with sliding local windows?

How does the DistAttn & load balancing work for global windows (i.e., each token attends to all tokens)?

How does the DistAtten & load balancing work for MQA models?

-. **Unclear Evaluation Setup**:

> Figure 3

Is it measured in a distributed setting or on a single GPU?

-. **Intertwined with Other Work**

> LIGHTSEQ + FSDP

FSDP replies a large batch/sequence size to amortize the weight AllGather overhead; how will an enlarged sequence length affect LIGHTSEQ's performance?

Does this work use FSDP for cross-node and LIGHTSEQ for intra-node communication?

How about we use only LIGHTSEQ for cross-node and intra-node communication?

Can LIGHTSEQ scale up to more nodes, without leveraging FSDP?

-. **Limited improvement**:

For sequence length < 8K, LightSEQ is slower than DeepSpeed-Ulysses by up to 0.5x.

It seems that this work is only evaluated on Llama models.

Why sequence length at 128K show 1.44x overlapped cased vs 1.x no communication in Figure 5?

**Questions:**

*. See above

---

> ### Author Response · Authors · 2023-11-20
> **Reply to Reviewer AsB4 (Part 1)**
>
> **Q1:** Overstatement on proposing sequence parallelism, “why sequence parallelism is proposed in this work, instead of enhanced by this work?”
>
> **A1:**  We didn’t claim we are the first on sequence parallelism. In the related work, we have clearly stated that [1] is among the first to propose sequence parallelism. We also discussed [2] as another previous work.
>
> [1] Shenggui Li, Fuzhao Xue, Yongbin Li, and Yang You. Sequence parallelism: Making 4D parallelism possible.
> [2] Vijay Anand Korthikanti, Jared Casper, Sangkug Lym, Lawrence McAfee, Michael Andersch, Mohammad Shoeybi, and Bryan Catanzaro. Reducing activation recomputation in large transformer models. Proceedings of Machine Learning and Systems, 5, 2023.
>
> **Q2:** Overstatement on flash attention “By default, FlashAttention uses recompute during backward pass. How does this become a contribution of this work?”
>
> **A2:** We did not claim it is our contribution. Our contribution is a new checkpointing strategy that utilizes this recomputation to avoid another recomputation at the HuggingFace gradient checkpointing level, not the recomputation inside the FlashAttention kernel.
>
> **Q3:** The optimal way to shift compute: “Besides shifting… kv8.”
>
> **A3:** Thanks for this important and interesting question. The criteria of optimality is to evenly assign workload to different workers. Assume there are P workers, then the number of units of work is (1+2+...+P) = (1+P) * P / 2. As long as the scheduling finishes in ceil[(1+P) / 2] time steps, it is optimal in terms of compute balancing. Using 8 workers as examples, if the scheduling finishes in ceil[(1+8)/2] = 5 time steps, it is optimal. While there are certainly other schedules to achieve this, we provide one of these.
>
> **Q4:** How does the DistAtten & load balancing work for MQA models?
>
> **A4:** DIstAttn distributes the sequence dimension, and the system does not change whether it is a single head (MQA) or multi-heads (MHA).
>
> **Q5:** Is Figure 3 measured in a distributed setting or on a single GPU?
>
> **A5:** We thank the reviewer for this question. It is measured in an A100 40GB. We have updated the manuscript to reflect this.
>
> **Q6:** How does DistAttn and load balance works for sliding local windows or global windows
>
> **A6:** We thank the reviewer for this important comment. The focus of current work is on exact attention, which is also described in the related work section. Incorporating LightSeq with a sparse attention pattern is a natural next step. In this answer, we give an initial idea on how to extend LightSeq with local and global windows.
>
> For local sliding windows, the workload is naturally (near) balanced, regardless of single directional or bidirectional attention. Thus, simply disregarding the attention logic to non-local workers suffices. For instance, in exact attention, worker 7 needs to compute attention to all other workers. If the sliding window has a number of tokens equal to that of one worker, then worker 7 only needs to attend to itself and tokens in worker 6. In other words, it only needs to fetch key and value from worker 6, and compute attention. In terms of implementation change, it just needs to change the end condition of the for loop (from looping worker 1 - worker 7 to looping only from worker 6 - worker 7).
>
> Global windows can be interpreted in two ways. We’re not sure which one corresponds to the reviewer’s question so we answer for both. The first interpretation is one special design of the sparse attention (e.g., global attention in Longformer), where there are a certain number of global tokens that all later tokens need to attend to, which are used to capture the global information. To adapt DistAttn to this, one easy way is to keep a replica of all the global tokens in each worker, which is simple and practical as otherwise, the global tokens will need to be all-gathered at each time step. One can also split the global tokens evenly onto all workers and use all-gather upon computation to further reduce the memory requirement. The second interpretation is the bidirectional modeling in the full attention situation (e.g., BERT). In this situation, the users can simply disable the load-balancing schema and use the vanilla DistAttn as illustrated in Figure 6 of the updated manuscript. We introduce the load-balancing only to handle the bubble issue caused by the causal modeling, so there’s no need to apply it if it’s bidirectional attention as there is no computation bubble in this case.

---

> > ### Author Response · Authors · 2023-11-20
> > **Reply to Reviewer AsB4 (Part 2)**
> >
> > **Q7:**  Intertwined with Other Work and the usage of FSDP, “LightSeq … FSDP”
> >
> > **A7:** We thank the reviewer for asking about the relationship between LightSeq and FSDP. LightSeq is completely orthogonal to FSDP, and by default can be used by itself to reduce the activation volume in long context training. In the paper, we use FSDP (cross-node) so that the system consumes similar memory for the model parameters, for a fair comparison with Megatron-LM. This is intended for a rigorous analysis. The author believes that this combination is not a weakness of the method, rather a strength that LightSeq can be flexibly combined with other memory reduction techniques. We have updated the method and experiment section to avoid confusion.
> >
> > It is possible to design a more fine-grained hybrid parallelism strategy on LightSeq sequence parallelism and FSDP. It is highly dependent on the model size. For smaller models, it may be better to do FSDP cross nodes because the volume is small. For larger models, it may be preferable to FSDP only for intra-node.
> >
> > Regarding the question on “whether LightSeq can scale up to more nodes without leveraging FSDP”, our answer is positive. In fact, FSDP only helps reduce the model memory, and actually negatively affects the system scalability. In explanation, the communication volume is the sum of that in FSDP and that in LightSeq. In our main experiment, we have demonstrated scalability with the combination of LightSeq and FSDP. Using LightSeq alone will give faster training speed due to less communication when the model weights can fit in a single device memory.
> >
> > **Q8:** For sequence length < 8K, LightSEQ is slower than DeepSpeed-Ulysses by up to 0.5x.
> >
> > **A8:** Because the current P2P implementation is not yet highly optimized, LightSeq may not achieve theoretical communication lower-bound in shorter sequence training. We have analyzed this in the discussion section and are actively working on the engineering side. However, the core value of LightSeq lies in longer sequence training, where we have shown significant speedup compared to both Megatron-LM and DeepSpeed-Ulysses. In addition, DeepSpeed-Ulysses is an open-source implementation by the time of submission.
> >
> > **Q9:** It seems that this work is only evaluated on Llama models.
> >
> > **A9:** This paper focuses on optimizing decoder-based LLMs with casual masks, which we believe are sufficiently important models and are of the strongest interest to the ML community. We pick Llama for its popularity, and further evaluate LightSeq on several variants of Llama that have different system characteristics. Other autoregressive Transformers (like GPT, MPT, of a similar size) will have similar system performance to Llama.
> >
> > **Q10:** Why sequence length at 128K show 1.44x overlapped cased vs 1.x no communication in Figure 5?
> >
> > **A10:** We would like to ask the reviewer to clarify the question: what does “1.x” mean here. In this figure, the wall clock time is normalized so that the no communication implementation has a ratio 1. And the overlapped scheme has 44% higher wall clock time, while a non-overlapped implementation has 105% higher wall clock time. This shows the effectiveness of our overlapping optimization.

---

> ### Author Response · Authors · 2023-11-21
> **We are more than happy to address your further comments**
>
> Dear reviewer AsB4,
>
> We would like to appreciate again on your precious time and helpful comments. We hope our response and updated paper has adequately addressed your concerns. And we are more than happy to address any of your further comments.
>
> Best,
>
> The authors

---

> > ### Author Response · Authors · 2023-11-22
> > **We kindly request the reviewer to participate in the discussion before the deadline.**
> >
> > Dear reviewer,
> >
> > We thank again for your previous time and insightful comments. As the deadline of the rebuttal period approaching, we appreciate your feedback on our response, and we're more than happy to address any other last-minute question.
> >
> > Happy Thanksgiving week,
> >
> > The authors

---

> > > ### Comment · Reviewer_AsB4 · 2023-11-23
> > >
> > > Thanks for the great effort of the authors on the rebuttal and clarifications. The rebuttal has addressed some of my concerns, but still missed evaluations on non-LLAMA models and better performance over DeepSpeed-Ulysses. So I slightly prefer to hold my score. Thanks again for the authors.

---

> > > > ### Author Response · Authors · 2023-11-23
> > > >
> > > > We appreciate the reviewers for letting us know we have successfully addressed most of your concerns in the previous review. Once again, we sincerely appreciate the time the reviewer has put in helping us improve.
> > > >
> > > > Firstly, we answer the remaining of your concerns:
> > > >
> > > > (1) DeepSpeed-Ulysses: **we have substantial better performance than DeepSpeed-Ulysses at long sequence length**, up to 1.87x for 512K. **Performance at long sequence is the core measurement of the problem**. Moreover, **DeepSpeed-Ulysses is clearly a concurrent work: the Arxiv paper is posted three days before the ICLR deadline**. The author believes a paper should not be rejected for not giving much better results than a concurrent work, in the region where the paper is not optimizing for.
> > > >
> > > > (2) non-Llama models: Firstly, the Llama family is the most popular models, and is representative enough. Secondly, **all Transformers have similar architectures and share the same bottleneck, i.e., the attention module**, and we show our method addresses it well. We are happy to supplement experiments with other models. However, we believe this can not be a reason to give us a negative score.
> > > >
> > > > **In conclusion, neither of these two concerns could cause a negative score.**

---

> > > > > ### Author Response · Authors · 2023-11-23
> > > > > **We kindly request the reviewer to raise the score to positive.**
> > > > >
> > > > > In addition, from the first round of conversation, we believe that the major concern that leads to a negative scoring is the overstatement. Given your kind response, we believe that this concern has been well addressed.
> > > > >
> > > > > We would like to know the reason/justification why the reviewer still votes for rejection. If there is no major reason, we kindly request the reviewer to raise the score to positive ones.
> > > > >
> > > > > Happy Thanksgiving,
> > > > >
> > > > > The authors

---

### Official Review · Reviewer_ZAxH · 2023-11-04

**Soundness:** 3 good
**Presentation:** 2 fair
**Contribution:** 2 fair
**Rating:** 6
**Confidence:** 3

**Summary:**

This paper introduces a technique, DISTATTN, to speed up training of causal transformer models on clusters of GPUs and embodies them in a framework called LightSeq.  The LightSeq framework, based upon FlashAttention-2, shows $2\times$ speedup over an updated Megatron-LM (MLSys 2023).

DISTATTN addresses the load imbalance that occurs when splitting up the input sequence among worker GPUs while training a causal language model.  Naively, splitting up the input sequence across GPUs causes later worker GPUs to perform more computation as they need to attend to more of the input sequence so perform more computations.  The idea in DISTATTN is to have some of the earlier workers perform computations that would normally be performed by later workers.   An additional optimization applied is to shift the point of checkpointing to avoid some recompilations in the backward pass.

Analytical analysis shows a reduction from $10dN$ to $3dN$ in total communication volume versus Megatron-LM (2023) making the approach a bit more suitable for clusters without high bandwidth (and expensive) interconnects.

**Strengths:**

The results show a significant reduction in training time.

The general ideas are clear -- use load balancing and overlap communication and computation.

Reducing communication volume should enable model parallel training on cheaper hardware.

**Weaknesses:**

The concept of load balancing and overlapping communication and computation in parallel systems is not novel on their own even if the details here might be.

Details of the specific load balancing technique could be explained more clearly.

While the paper shows longer sequence lengths can be supported there were no accuracy or quality results demonstrating how much improvement the longer sequence length translated to.

**Questions:**

For right side of Figure 1, are all circles independent computations?  Otherwise it would be unclear why bubble size is only 4 after balancing.  For the rightmost figure it is unclear which computation have been moved to worker 1-3  for the upper triangle versus before balancing.   Perhaps the color scheme could be used to indicate this.

I found it unclear the relationship between Figure 1 and 2 unclear.  In Figure 1 it looks like worker 7 should be computing attn(q7,k1v1) in the first timestep, but in Figure 2 in the first timestep what is shown is attn(q7,k7v7).  Similarly, the communication pattern in Figure 2 is a bit unclear.  Is there a simple formula for what attn query/key/value combination is computed on a worker in a given timestep and what values are  communicated from/to a given worker on a given timestep?

On Page 5 it is mentioned that the evaluation uses variously 2 A100 DGXs and an 2x8xA100 in house platform without infiniband.  However, as both of these have 2x8 GPUs, it is unclear which system the measurements are for in Table 1.   I would expect given the reduction in communication volume that a "cheaper" cluster without NVLINK might give decent results and was interested to see results for a comparison vs. DGX, but it seems like they are missing from the paper and supplemental.

What is the speedup versus FlashAttention-2?  If I understood correctly for LightSeq you started with the FlashAttention-2 framework, so I would expect to see speedups with respect to that system too.

The longer sequence lengths explored in Table 2 and 3 should yield better accuracy / quality on the tasks the networks are applied to, but  there appeared to be no results demonstrating this.

---

> ### Author Response · Authors · 2023-11-20
> **Reply to Reviewer ZAxH**
>
> We thank the reviewer for the insightful and helpful feedback! We would like to address your questions in the below response.
>
> **Q1**: The concept of load balancing and overlapping communication and computation in parallel systems is not novel on their own even if the details here might be.
>
> **A1**:  We agree with the reviewers that load balancing and overlapping communication has always been important topics in parallel systems. However, we want to point out LightSeq is the first work that discusses how to design and implement these two techniques in the space of sequence parallelism.
>
> **Q2**: Details of the specific load balancing technique could be explained more clearly.
>
> **A2**: We thank the reviewer for raising this clarification question. We added a section in the appendix (Appendix C) to illustrate the load-balancing design. Table 6 shows the computation and communication pattern before balancing, while Table 7 shows the pattern after load-balancing. We will also answer your question in the next paragraph and please feel free to let us know if there is still anything unclear to you.
>
> Each circle is a unit of computation. Circles in the same color mean that they are computed in the same time step.  For instance, the rightmost and bottommost circle means that at time step 1 (t1), worker 8 is executing attn(q8, k8, v8). Similarly, green color denotes computations that happen at the second time step (t2). At t2, worker 1 is executing attn(q8, k1, v1). We have updated the figure and the caption to communicate the idea better.
>
> **Q3**: The relationship between Figure 1 and Figure 2 is unclear - “I found it … on a given timestep”?
>
> **A3**: Following up on the previous answer, at the first step (blue color), worker 7 is computing the circle corresponding to q7 and kv7, i.e. attn(q7, k7, v7).
>
> Yes, we would like to provide a formula here. (Before rescheduling) the computation a worker w at time t is $attn(q_w, k_{(w-t+1)}, v_{(w-t+1)})$, i.e. in a backward manner - worker 7 computes attn(q7, k7, v7) on time 1, and attn(q7, k6, v6) on time 2. After rescheduling, if a worker w helps other workers (e.g. worker 1, 2, 3 in Figure 1), it is computing $attn(q_{(w+P-t+1)}, k_w, v_w)$, i.e. also in a backward manner - worker 1 computes attn(q8, k1, v1) on time 2, attn(q7, k2, v2) on time 3. For workers who do not help others, the computation remains the same (but the maximal time step gets reduced because of load balancing).
>
> For communication, the worker pre-fetches the values needed for the next time step. Before rescheduling, a worker w at time t is computing $attn(q_w, k_{(w-t+1)}, v_{(w-t+1)})$. Thus, it will prefetch $k_{(w-t+1)}, v_{(w-t+1)}$ at time (t-1). In other words, at time t, worker w prefetches $k_{(w-t)}$ and $v_{(w-t)}$. As an example, worker 7 computes attn(q7,k6, v6) in time 2, so it prefetches k6 and v6 in time 1. With rescheduling, the logic is very similar. If a worker w helps other workers, it will compute $attn(q_{(w+P-t+1)}, k_w, v_w)$ at time t, and thus prefetches $q_{(w+P-t+1)}$ at time t-1. Additionally, it sends the output o back when it is ready. For workers who do not help others, the prefetching logic for key and value is the same.
>
> **Q4**: The configuration of Table 1: “On page 5 it is mentioned that… and supplementary”.
>
> **A4**: Table 1 is using the second cluster setup (2 DGX boxes). We found in the previous We have updated the manuscript to indicate that the default setup is the 2 DGX boxes, and thanks again for the helpful feedback.
>
> To test the performance of inter-node training, we mainly report results on DGX boxes with Infiniband. This is because the current implementation of the P2P component is not yet highly optimized (as we brought up in the discussion section),). We are actively working on the engineering side to improve this. To be more specific, in higher bandwidth inter-node training settings, our communication overlapping component results in a very noticeable speedup, and we thus use the 2 DGX boxes as the default testbed. We have also updated the discussion of the manuscript to make our implementation roadmap clearer.
>
> **Q5**: Speedup against FlashAttention: “What is the speedup.. to that system too”.
>
> **A5**: LightSeq and Flash attention are orthogonal work, one is distributed and the other is on a single GPU. Conceptually, LightSeq reduces to Flash attention when P=1. When scaling to P=8, we have observed 7.5x speedup compared to flash attention (Figure 5 Left), a near-linear scaling.
>
> **Q6**: No accuracy report for longer sequences: “The longer … demonstrating this”.
>
> **A6**: In this work, we focus on the system/infrastructure side of long-context training. However, the gain of longer sequences training has been validated in several previous works, e.g. in [1], [2].
>
> [1] FlashAttention: Fast and Memory-Efficient Exact Attention with IO-Awareness
>
> [2] Longformer: The Long-Document Transformer

---

> ### Author Response · Authors · 2023-11-21
> **Thank you and please let us know any further comments!**
>
> Dear Reviewer ZAxH,
>
> We are deeply encouraged by your helpful and positive feedback, and sincerely appreciate your efforts. We hope our answer and correspondingly updated manuscript makes the paper clearly and stronger. Please let us know if you have any further comments, and we are more than happy to address them.
>
> Best,
>
> The authors

---

> > ### Author Response · Authors · 2023-11-22
> > **We kindly request the reviewer to participate in the discussion before the deadline.**
> >
> > Dear reviewer,
> >
> > We thank again for your previous time and insightful comments. As the deadline of the rebuttal period approaching, we appreciate your feedback on our response, and we're more than happy to address any other last-minute question.
> >
> > Happy Thanksgiving week,
> >
> > The authors

---

### Official Review · Reviewer_Fg9E · 2023-11-10

**Soundness:** 2 fair
**Presentation:** 2 fair
**Contribution:** 2 fair
**Rating:** 5
**Confidence:** 3

**Summary:**

LightSeq introduces a novel approach for training large language models (LLMs) with extended context lengths, overcoming the limitations of previous model-parallel systems like Megatron-LM. Unlike Megatron-LM, which is limited by the number of attention heads and incurs high communication volumes, LightSeq efficiently partitions over the sequence dimension. This makes it adaptable to various model architectures and significantly reduces communication needs by up to 4.7×, while also overlapping communication with computation. In comprehensive experiments, including on the Llama-7B model, LightSeq demonstrates up to 2.01× speedup in end-to-end training and supports up to 8× longer sequence lengths for models with fewer heads compared to Megatron-LM.

**Strengths:**

* Efficient training of long-sequence models is an important goal.
* Demonstrates runtime speedup over Megatron-LM for long sequence lengths.

**Weaknesses:**

* Leveraging sequence-level parallelism in transformer training (Section 3.1) is not novel (e.g., https://arxiv.org/abs/2105.13120).
* Overlapping computation and communication (Section 3.2) appears to be a fairly standard technique and may not be as effective on more recent GPUs like the H100, which have significantly higher compute capabilities than the A100, not to mention optimizations such as transformer engine, tensor memory accelerator, etc.
* The configurations for parallelism are questionable.

**Questions:**

Thank you for submitting to ICLR 2024. Leveraging sequence-level parallelism is an interesting direction to scale both sequence length and training efficiency. However, I have the following questions:

* This is not the first work to leverage sequence-level parallelism for training transformer models. For example, how does the proposed work compare, both qualitatively and quantitatively, to [this 4D sequence parallelism work](https://arxiv.org/abs/2105.13120)?
* The parallelism configurations are unclear. What is the batch size used for training? If the batch size is high enough, wouldn't a combination of tensor/data/pipeline parallelism provide sufficient parallelism to achieve high utilization of GPUs, even without utilizing sequence parallelism?
* It is somewhat odd to take results from different clusters for different experiments (e.g., the results in Table 2 were drawn from a small memory configuration without InfiniBand). Is there a convincing reason for this?
* In Table 3, the authors used a batch size of 1, which does not seem to be a reasonable configuration for training. Can you elaborate on this choice?
* Regarding the explanation of pipeline parallelism, the authors attribute the low performance to “high memory pressure in the first stage.” Couldn't this be due to poor partitioning between stages?

---

> ### Author Response · Authors · 2023-11-20
> **Reply to Reviewer Fg9E**
>
> We thank the reviewer for the insightful and helpful feedback! We would like to address your questions in the below response.
>
> **Q1**: How does the proposed work compare, both qualitatively and quantitatively, to this 4D sequence parallelism work?
>
> **A1**: We discussed 4D sequence parallelism in the related work section:
>
> >“[1] is among the first to parallelize along the sequence dimension. However, it is not optimized for the computational pattern of causal language modeling and is incompatible with memory-efficient attention, which is crucial to long-context LLM training. ”
>
> In other words, the 4D sequence parallelism work is the first to propose this concept but it’s not optimized for modern LLM training and, thus is unable to support very long sequences efficiently. Quantitatively, LightSeq supports at least 8x longer sequences than this work, and achieves 4.45x - 5.64x speedup. We have added this experiment results and further analysis in the updated paper (appendix: Comparison with Ring Self-Attention). We provided the main experiment numbers below.
>
> ### Maximal sequence length on Llama-7B on the DGX (A100-80GB) cluster
>  |           | 1 Node (8 GPUs) | 2 Nodes (16 GPUs) |
> |-----------|-----------------|-------------------|
> | RSA       | 32K             | 64K               |
> | LightSeq   | > 256K          | > 512K            |
>
> ### Per iteration time comparison with RSA (seconds) on the DGX clusters
>
> |           | 1 Node (32K sequence length) | 2 Nodes (64K sequence length) |
> |-----------|--------------|---------------|
> | RSA       | 14.10        | 30.49         |
> | LightSeq   | 2.50         | 6.85          |
> | Speedup   | 5.64x        | 4.45x         |
>
>
> **Q2**: The parallelism configurations are unclear. What is the batch size used for training? If the batch size is high enough, wouldn't a combination of tensor/data/pipeline parallelism provide sufficient parallelism to achieve high utilization of GPUs, even without utilizing sequence parallelism? In Table 3, the authors used a batch size of 1, which does not seem to be a reasonable configuration for training. Can you elaborate on this choice?
>
> **A2**:
> When combined with tensor/data/pipeline parallelism, LightSeq sequence parallelism can be thought of as a better parallelism approach *replacing TP* to support long context. To see this, we note the LLM training memory utilization can be decomposed into two parts. The first part is related to the model parameters, i.e., memory used to store parameters, gradients, and optimizer states. The second part is related to the activation. It is worth noting that the memory for activation dominates when increasing the context length and the batch size only affects the memory for activation. Given this premise, we know that, firstly, data parallelism wouldn’t reduce either part, so we could consider it as orthogonal and only discuss the DP=1 case. Secondly, in Table 2, we show that the combination of model parallelism and pipeline parallelism handles shorter sequence lengths in a variety of scenarios, even when batch_size=1. This indicates that our sequence parallelism can support a very long context, as it would avoid TP+PP running out of memory.. In addition, Table 2 shows that using TP alone can effectively support the same length as LightSeq, but it requires more communication and takes longer than LightSeq as shown in Table 1. Increasing the batch size has the same effect on TP and LightSeq sequence parallelism (LightSeq is used with FSDP in experiments). Therefore, LightSeq should be a better choice than TP in supporting long context lengths.
>
> **Q3**: Regarding the explanation of pipeline parallelism, the authors attribute the low performance to “high memory pressure in the first stage.” Couldn't this be due to poor partitioning between stages?
>
> **A3**: **We believe optimizing pipeline partitioning may not solve the problem**. We follow the standard practice in Megatron-LM to evenly partition the layers onto different devices and utilize micro-batches to reduce the bubble size. This partitioning strategy prefers to have an even workload across stages. If we partition it into uneven stages to balance the memory, it will cause unbalanced computation (stragglers), leading to inefficiency.
>
> [1] Shenggui Li, Fuzhao Xue, Yongbin Li, and Yang You. Sequence parallelism: Making 4d parallelism possible. arXiv preprint arXiv:2105.13120, 2021.

---

> ### Author Response · Authors · 2023-11-21
> **We are more than happy to address your further comments**
>
> Dear reviewer Fg9E,
>
> Once again we sincerely thank your precious time and helpful comments. We hope our response and updated paper has adequately addressed your concerns. And we are more than happy to address any of your further comments.
>
> Best,
>
> The authors

---

> > ### Author Response · Authors · 2023-11-22
> > **We kindly request the reviewer to participate in the discussion before the deadline.**
> >
> > Dear reviewer,
> >
> > We thank again for your previous time and insightful comments. As the deadline of the rebuttal period approaching, we appreciate your feedback on our response, and we're more than happy to address any other last-minute question.
> >
> > Happy Thanksgiving week,
> >
> > The authors

---

### Author Response · Authors · 2023-11-20
**General Response**

We thank all reviewers for their constructive feedback! We are encouraged that reviewers appreciate our work for the following reasons:
1. The problem of efficient long-sequence training is important.
2. The general idea of load balancing and overlapping communication and computation is clear.
3. Demonstrate a significant training time reduction.
4. Provide an open-source implementation for reproducibility.

We made the following revisions in the updated version (highlighted in blue) to address the common questions raised by reviewers:
1. We added an experimental comparison with 4D parallelism (Ring Self-Attention) in Appendix D which shows that LightSeq supports a longer sequence length than Ring Self-Attention, i.e., at least 8x longer for both the intra-node setting and the inter-node setting. We also show that LightSeq has a faster training speed than the Ring Self-Attention, i.e., 5.64x faster for the intra-node setting, and 4.45x faster for the inter-node setting, respectively.
2. We added one section in the appendix (App. C) with two figures detailing the computation and communication pattern of DistAttn in the 8-worker scenario. Specifically, Figure 6 shows DistAttn design before applying load-balancing and Figure 7 shows DistAttn after applying load-balancing. The communication schedule is also reflected in both figures by comparing the tensors stored by each worker at two consecutive time steps.
3. We reorganized the contents according to the reviewers’ suggestions. Specifically, we moved the discussion of comparison with DeepSpeed-Ulysses to the experiment section and the discussion of the pipeline parallelism to the related work sectionto improve the paper flow.
4. We clarified some experimental settings that were confusing.
5. We added a discussion on the unbalanced memory usage in pipeline parallelism in the related work section and added an experimental illustration of this phenomenon.

---

### Meta-Review · Area_Chair_n3Y3 · 2023-12-04

**Metareview:**

Reviewers find this paper to be on borderline. While the paper combines some existing ideas in distributed systems implementation to cut down computational costs for long seq transformers, reviewers find limited novelty as the paper is mostly combining existing techniques. Reviewers overall find contributions to be limited and do not suggest acceptance. Hence I recommend rejection.

Note that comments from Reviewer AsB4 asking for comparisons to a parallel work are ignored for this decision.

**Justification For Why Not Higher Score:**

Limited novelty.

**Justification For Why Not Lower Score:**

N/A

---

### Decision · Program_Chairs · 2024-01-16

Reject